# Cartography and Art: A Comparative Study Based on Color

**Leda Stamou**

Cartography Laboratory, School of Rural, Surveying and Geoinformatics Engineering, National Technical University of Athens, 9 Iroon Polytechniou, 15780 Zografou, Greece; lestamou@central.ntua.gr

**Abstract:** Color occupies a prominent place in the bibliography of cartography, as it is an important element in the formation of cartographic symbolization. Apart from the technical issues of its application to maps, color theory is one of the elements that connect maps with art. In this paper various cartographic trends and their origins are examined and correlated with the artistic periods in which they were developed in order to investigate and document the extent to which maps follow the artistic movements and, particularly in the art of painting, concerning the form and the content of the maps and whether color can be used as an identification element of the art trend and the corresponding period. The research spans from the end of the Middle Ages to the 21st century and is referred spatially in Western Europe, including Italy. The comparison of colors is made in both descriptive and quantitative terms through the commentary of hue, brightness, and saturation, as well as through plotting them in the color wheel, a process that allows an overview of the range and location of color sequences. Concluding, the paintings and maps that were selected and examined in detail support the effect of painting on maps, without implying that it is intentional.

**Keywords:** color; color theory; color schemes analysis; visual variables; cartographic symbolization

## 1. Introduction

In the evolution of cartography, remarkable relations between cartography and art appear, not only during those periods when maps were created by highly skilled engravers and artists, but also later. Even during the mature period of cartography since the last decades of the 20th century, the map composition is widely related to the appearance of the map, both in terms of the symbols used, the layout, and the overall visual impression. Despite the fact that, nowadays, cartography has been completely dominated by science and technology, without the artistic skills required as in earlier periods of the history of cartography, during the phase of cartographic composition, the cartographer is called upon to be imaginative and creative in order to support the "good map design". It is quite clear that the theory of cartography, along with the cartographic tradition and practice, allow the cartographer much less freedom than art allows to the artist, provided that cartographic design challenges the cartographer to find effective solutions to graphic design issues. The issue of the relationship between cartography and art is one of the issues that concern the cartographic community. The ICA "Art and Cartography Commission" has a significant role in promoting creative research and scholarly publications on art and cartography in all of its aspects, for both the academic audience and the general public.

Cartography has benefited greatly from science, as far as the cognitive areas of cartographic representation, communication, color vision theories, optics, color theory, visual perception, and psychology are concerned. Art has also. Many artistic movements, such as Impressionism, Neo-Impressionism, Pointillism, Fauvism, and Modernism, were based on the scientific knowledge that emerged from the theory of color, the trichromatic theory, and the opponent process theory. It should also be noted that contemporary art shares additional characteristics with science: modern artists, having rejected aesthetics as the defining feature of their work, have based art on a practice that can be imaginative, creative, provocative, and exploratory [1].

In addition, part of exploring the relationship between cartography and art focuses on the use of color [2,3]. Paintings can be a source of inspiration for cartographers when it comes to choosing and using color. Color, due to its inherent characteristics is perhaps the most complex and multifaceted element of the cartographic design. Painters have highlighted color and form as the two dominant elements of their work, as evidenced by the study of art history [4]. In particular, color has been a central element in various artistic movements [4] and, in fact, is often enough to determine the artistic period of a painting and often the painter.

Another notable issue that frequently arises in the cartographic bibliography is the fact that maps inspire and stimulate the interest of artists, being the subject or medium for artistic creations. Great Dutch painters such as Vermeer, Pieter de Hooch, and many others have portrayed maps in their paintings during the exploration period, emphasizing this way the important role played by this medium, as it portrayed the new territories. Johannes Vermeer in particular, portrayed maps in ten of his paintings [5]. For example, the map made by Balthasar Florisz van Berckenrode and published by Willem Jansz Blaeu in 1621, is portrayed in the "Officer and laughing girl" (Figure 1, adapted from [6], Attribution: Johannes Vermeer, Public domain, via Wikimedia Commons) and also in the "Woman in blue, reading a letter" [5].

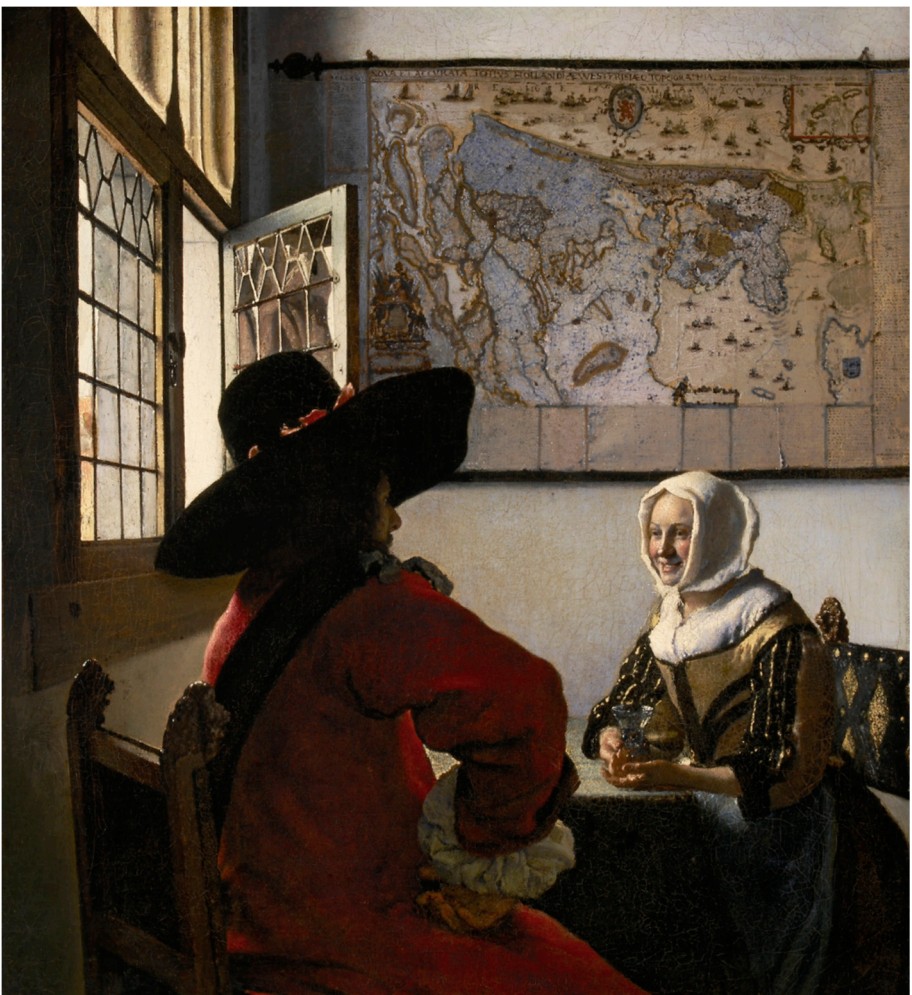

**Figure 1.** Johannes Vermeer: Officer and the laughing girl.

The Surrealists published their world map in 1929, which they intentionally distorted by formulating a political message against Imperialism and the conventional ethnocentric representation of the world [7,8]. It was published in a special issue of the Belgian

magazine Variétés on "Le Surréalisme en 1929" in June 1929. Respectively, Situationists drew the map "The Naked City", which shows how this group of artists and activists perceived the neighborhoods as they wandered around the city, with the aim of providing alternative ways of urban space, and the crossing of psychogeographical boundaries [9,10]. Jasper Johns (b. 1930), has created paintings and collages portraying the map of the USA using bright colors and vivid touches of abstract expressionism [11]. So maps are often a source of inspiration for many artists. Joyce Kosloff and Mona Hatoum utilize maps to format their iconic artistic works [12,13]. From the anthropomorphic maps of Bunting and Münster [14,15] in the 16th century to "Carto-morphic" people in the 21st century: Matthew Cusick's collages of cartographic people and landscapes in the "Map Works" collection [16] are part of the cartography and art dialogue. This dialogue transcends the boundaries of the visual arts and in several cases infiltrates in literature, a fact indicative of the power of the map. The case of Rebecca Solnit and her book "Unfathomable City: A New Orleans Atlas" is typical. There, she uses a performance map of the concentration of lead in the soil, superimposing texts with political lies from 1699 that influenced the social and racial development of the city [9,17].

One could claim that the common elements of art and cartography are representation and symbolization. In art, the representation comes to light via the expressive means of the artist, while in cartography via the skills and knowledge of the cartographer, that is, the method of symbolization, the symbols themselves, and the structure of the map. The aggregation of the individual symbols of a map constitutes the result of its objective, which is usually the geographical reality. As an inevitable stage of cartographic composition, symbolization could be defined as the graphical encoding of information, or, in particular, the use of visual variables to represent data aggregates resulting from categorization, simplification, and amplification [18].

A map is a graphic representation of geographical reality, which is implemented through cartographic symbolization. Unlike the image characterized by the property of similarity, the map emerges as a result of an abstract process: the entities to be represented are selected and symbols with conventional relation to the signifier are used. Arthur Robinson emphasizes cartographic representation as: "The principal task of cartography is to communicate environmental information ... The task of the map designer is to enhance the map user's ability to retrieve information" [19] (p. 17). The symbols are determined by cartographic rules and design constraints. In the course of information from the geographical reality to the user through the map, the role of the cartographer is crucial as he/she implements the cartographic abstraction that concerns both the selection of the information to be displayed and the selection and configuration of the symbolization. The cartographic rendering of spatial information is a process of transformation from its physical form to a form of non-literal symbolization contained in the map. The visual perception, the mental processing, and then the recognition and/or interpretation of these symbols by the user of the map are critical components in achieving the goal of its creation, i.e., in the transmission of the geographical information. If cognitive and geometric accuracy are taken for granted, based on our knowledge of the world and the available technological means, the main task of the map composition is the selection of appropriate symbols and their organization in a functional communication dipole between the cartographer and the map user, using the symbol as a carrier.

Color and its application to the individual cartographic elements is a critical factor, not only because it is one of the elements of symbols formation, but also because it establishes harmony, balance, contrast, visual hierarchy, and image-background organization. In the image formed by the map, what will pop-up, what will recede, what will be clear, and what will be blurred, bright or dark, what will be different or similar, depends mainly on the color [20]. From an artistic point of view, color is a powerful means of artistic expression and activates the feelings of the observer. It is remarkable that three of Bertin's visual variables are related to color: hue, brightness, and saturation [21]. Indicatory of its distinguished role in cartographic design, cartographic symbolization, and cartographic

communication is the fact that every single cartographic textbook written by prominent cartographers and scholars, devotes a special section to color description, analysis, and perception, as well as color models and color spaces [19,22–24].

Socio-political changes and technological progress have had a catalytic effect on both the acquisition of knowledge about the geographical area and the means of cartographic expression. This study is an attempt to investigate and document the influence of art—and particularly the art of painting—on the form and the content of the maps, with emphasis to the use and the role of color. The comparison is based on the change of the subject, as well as the means of expression in paintings and maps, over time. The form of maps, paintings, and other graphic works is also the subject of commentary, as well as the infiltration of the dominant artistic point of view in cartographic design. To some extent, this approach is based on historical, artistic, and cartographic elements. The aim of this study is to investigate whether and to what extent maps are influenced by the prevailing artistic trend of the time and whether this can be exploited today.

Based on the way color is used in various artistic movements, the aesthetic and artistic relationship between paintings and maps is sought, that is, whether and to what extent the paintings infiltrate the works of cartographers and, in some cases, vice versa. Thus, an attempt is made to examine and/or correlate the parallel evolution of the artistic and cartographic periods. For this reason, maps are examined based on the artistic period. For this research, maps and paintings related to space and time are sought. The choice of maps examined in this study does not constitute historical documentation, but are considered as an anthology of those that signal the change in the format and content of the maps, in the context of the history of cartography.

## 2. Methodology

Before describing the methodology, a reference to the selection of maps and paintings is necessary. The reasonable question raised is "which maps compare with which paintings". Based on the history of cartography, cartographic schools in Europe were used as the fundamental criterion. Maps of well-known Italian, Spanish, Portuguese, German, Dutch, French, and English cartographers defined the framework as well as the timeline. For the selected cartographic periods, from the medieval European world maps ("mappae mundi") to the Italian and Portuguese portolans, the world maps and atlases of the Age of Exploration to the maps produced by scientific and technical developments, even up to the online map services, the corresponding artistic period was examined, with respect to their general characteristics and the use of color. Thus, the history of cartography is linked to the history of art and this resulted in a visual examination and comparison between maps and paintings of the same period. As the original works (maps and paintings) are not accessible, the study is based on the search for the corresponding digital images and the selection was made through/from official national digital libraries, museum websites, and collectors' websites. For each map examined, paintings from the same period are selected and common formatting elements are sought. The colors, the structure, and the margin of the map try to find their counterparts in the paintings.

The methodology followed in this research has two phases: the qualitative approach and the quantitative one. During the qualitative phase, the comparison of the colors is initially descriptive, i.e., their visual characteristics are examined based on the visual variables related to color: hue, brightness (value), and saturation (chroma) [25]. It is examined which hues have been used, which part of the color wheel they cover, and what is the variation of brightness and saturation. At the same time, the role of color as a means of expression, as well as the way it is used to distinguish spatial information, is identified, as described in the next sections. Thus, the qualitative phase concludes to a verbal description of the colors used. In order to visualize the results of the verbal description and comparison, as well as to highlight the affinities of the colors, a quantitative phase has been adopted to support and document the results with metrics concerning color coordinates, as described below.

The quantification phase is then implemented, using a combination of ColorSchemer Studio 2 and Adobe Photoshop software (depending on the quality of the image), so that color schemes of maps and paintings are created and plotted in the color wheel as shown in Figure 2, which is a typical example of color analysis. In this case, the selected colors correspond to the colors of line and area symbols of an examined map. In ColorSchemer Studio 2, the colors can be selected from the image, either automatically or manually, to form the image's color scheme, which is portrayed as a set of color patches. The selected colors are then automatically transferred to the color wheel using their RGB primaries (identified by the software) and they are examined on a case-by-case basis for their perceptional attributes (hue, brightness, saturation), as well as the affinities of the colors. This transfer can also be made by manually deriving colors from an image through a different application (e.g., Adobe Photoshop). The color schemes can then be exported to standalone image files of color patches or can be combined with the source image for presentation purposes (see to the right of Figures 4, 5, and 10). Additionally, the color schemes can be exported to various formats (e.g., html, css, etc.) for any further use. Thus, lists of coordinates, either in RGB or HEX, can be used for numerical presentation, as shown in Table 1. It is noted that the color "coding" in Figure 2 and Table 1 are used here for reference purposes.

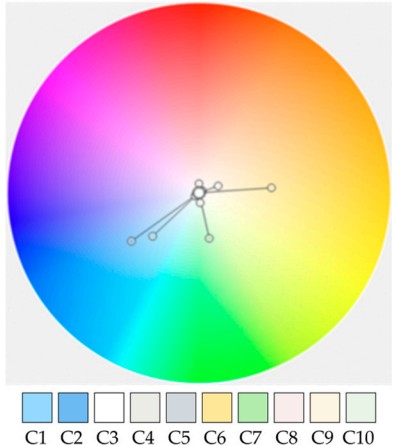

**Figure 2.** Color analysis, typical example.

**Table 1.** Color coordinates, typical example.

| Color | HEX | RGB |
|---|---|---|
| C1 | #AADAFF | 170,218,255 |
| C2 | #89BCF3 | 137,188,243 |
| C3 | #FFFFFF | 255,255,255 |
| C4 | #EDEBE8 | 237,235,232 |
| C5 | #D7DADD | 215,218,221 |
| C6 | #FDF4E2 | 253,244,226 |
| C7 | #C3ECB2 | 195,236,178 |
| C8 | #F9EDED | 249,237,237 |
| C9 | #FFE99E | 255,233,158 |
| C10 | #ECF7EA | 236,247,234 |

In Figure 2, the color scheme of the image is shown on the color wheel, and thus, each color is visually arranged in relation to the set of colors displayed on the screen. This kind of visualization supports conclusions about the range of colors used, which area of the color wheel they occupy and their brightness (or saturation, depending on the choice). Especially for the brightness (or saturation), it is visualized and assessed by the distance of the color from the center of the color wheel (Figure 2), while numerically it is expressed through the corresponding HSB (Hue-Saturation-Brightness) built-in tool.

Due to the use of digital images, the issue of color accuracy in relation to the original work (map or painting) is raised. This is uncontrollable in quantitative terms. For this reason, the research is made using more than one source, in order to cross-reference the data, and it is preferred that the selection be made from official organizations and institutes that demonstrate the required dependability. This is one of the reasons why the numerical description of the colors is not emphasized, but to a considerable extent the main weight is given to the verbal and visual (via the color wheel) description. Nevertheless, the numerical results (color coordinates) could be further used, for example, as a base in cartographic compositions and symbols creation. Any color profiles attached to digital images are retained during processing.

Due to possible Creative Commons restrictions, most of the images used are not included in this article, but the relevant source links are provided. More than 100 maps and paintings were compared and analyzed as part of an extended research [20], for the period from the end of the Middle Ages to the beginning of the 21st century, in Western Europe. It is pointed out that many more had been visually examined prior to the final selection. This paper presents a limited number of them, that is indicative and on the same time representative for each period. Results and discussion are presented in the following paragraphs.

## 3. Analysis and Results

### 3.1. Gothic Period

During the Middle Ages (from the 5th to the 15th century A.C.), maps, known as mappae mundi, reflect the influence of the Church: the cartographic representation is allegorical, as is the pictorial representation in images and murals. Typical examples are the map of Ebstorf (Germany, c. 1234) and the map of Hereford (United Kingdom, c. 1285). They are oriented with the East at the top, and Jerusalem is located in the center. The perception and description of the world revolves around the teachings of religion and this is also expressed cartographically. Maps of this era are characterized as "pictorial" [9], but it is easy to recognize the influence of early narrative painting. This artistic term describes an art form that "tells a story" and as general as it sounds, it is literal. It can describe a moment, an event, or a series of events. In early narrative painting the perspective, the actual sizes, and the relative positions are not as important as the transmission of information in the form of narrative. Typical examples of such paintings exist from the beginning of the history of painting until today, not only with the works of the so-called Naive painters, but also with works of historical and mythological content during the Renaissance, Baroque, etc. Note that one of the modern trends in cartography is the creation of storytelling maps utilizing software applications. This way of depicting landscapes, people, and events has left many samples in cartography and especially in the Middle Ages. The "mappae mundi" are a typical example, but the influence of this particular perception and performance of the space, gave important samples in other categories of maps, such as the portolans. This category of maps was created at the end of the 13th century, initially in Italy, and more specifically in Genoa and Venice (followed by Spain and Portugal), and formed the basis of the maps of the time of the discoveries (15th century). Portolans portray the shoreline and compass directions, typical names associated with the coast, ports and capes, while the interior of the coast is either left empty or decorated with symbols of power and compass roses that are depicted in characteristic positions. Toponyms along the coastal points of interest are written on the inner side and with a direction perpendicular to the coastline.

The oldest surviving portolan specimen is the Carte Pisane (c. 1292) and is attributed either to an unknown cartographer or to the Italian cartographer Pietro Vesconte (1310–1330). Contemporary of the Italian School of cartography is the School of Majorca, staffed by prominent Jewish cartographers, cosmographers, and instrument makers, as well as some Christian collaborators, which flourished from the 13th to the 15th century until the Spanish Holy Examination. This school also includes those who were active in Catalonia.

Vesconte's portolans are characterized not only by the religious themes in the marginal illuminations, but also by the influence of painters, such as Giotto di Bondone (1267–1337). Both the structure and the colors used in his portolans look like those used in Giotto's paintings. Inspecting one of his maps made in Venice at 1318 [26], (p. 21), the resemblance to Giotto's paintings is very impressive: the map is located in the center and the corner illuminations depict religious themes (i.e., the Annunciation of Virgin Mary), facing the map, exactly like the way peripherals are looking at the central person in the paintings of religious content. The forms on the map and in the painting are similar, since the era is characterized by a specific style based on the use of specific dyeing materials and dyeing methods for the depiction of leather, fabrics, and folds of clothing. The colors have a great resemblance: gold background, beige, green, red-tile, and blue, most of them located in the area of warm colors. Green (light and dark), tile, ocher, and gold are used not only in the decorative elements of the map, but also on the map itself. This particular cartographer maintains the same structure in his works, with the presence of persons or symbols of religious authority. The golden background, which is the backdrop to the religious scenes acting as a curtain separating the physical from the spiritual world, is also found in the maps of Vesconte, as well as in the paintings of the time (e.g., Giotto di Bondone: "The Madonna Di Ognissanti", c. 1310 [27], Duccio di Buoninsegna: "The Calling of the Apostles Peter and Andrew", c. 1308–1311 [28], and many others).

### 3.2. Cartography in the Renaissance or the Renaissance of Cartography?

The nautical atlas of Battista Agnese (compiled in 1544) [29] is compared to Titian's "Bacchus and Ariadne" (1523) [30] and the results are portrayed in Figure 3. Both color schemes are located to the same partition of the color wheel, most of them covering the sector from red to yellow.

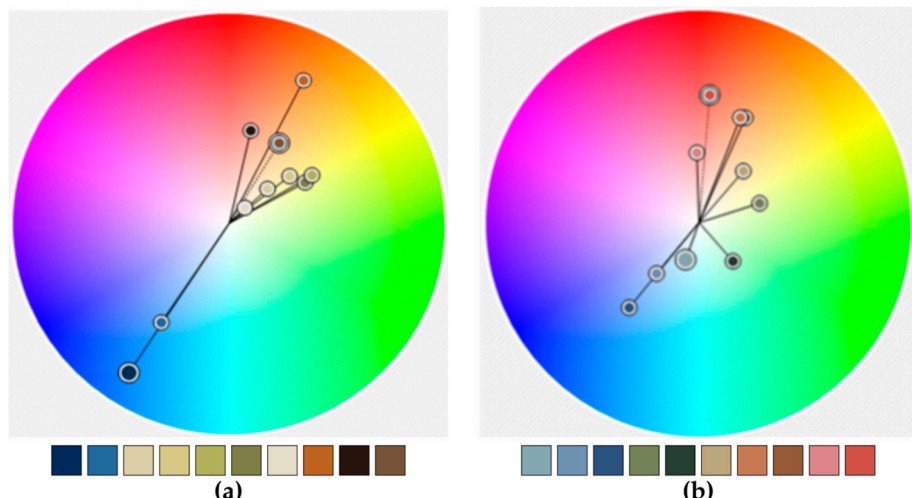

**(a)**         **(b)**

**Figure 3.** Color analysis and comparison: (**a**) Battista Agnese nautical atlas (**b**) Titian painting.

Angelo Freducci's portolans are characterized by vivid colors: blue, green, red, ocher, green-gray, the design of colorful compass roses, and the use of different ways of coloring the islands. On his map of the Caribbean for example (c. 1550) [31], the small islands are filled with solid colors, the middle ones have a colored border that follows the interior of the coastline and fill background with the same color in a lighter tone, while on the large islands the coastline is reinforced with a colored border to the interior of the island. The color impression produced by this map is very close to that of the painting by Domenico Beccafumi "The Holy Family with Angels", c. 1545/1550 [32]. In both cases, the color is used to highlight the important: the islands and the cruising directions in the case of the map and the central face in the painting, while the remaining elements or faces are lost in the neutral background. The visual characteristics in most of his maps are similar and it

is notable that some of them carry a very strong pictorial character, especially in terms of cities and mountains symbolization. The mountains are represented by triangular shapes and, in color, by the imitation of chiaroscuro, with the light tones to the east where the light source is placed and the dark tones to the west. Blue is used for rivers, green for lowlands, beige and brown for cities, and almost orange for mountains. Remarkable color similarity is noted between this map and the painting "The Miraculous Draft of Fishes" (1545) by the Venetian painter Jacopo Bassano [33]. From the comparison of the color analysis, it is obvious that the color sequences are placed in the same part of the color wheel, in the area of the warm ones, with small differences in saturation.

Worth mentioning is the Vatican "Maps Gallery" (Galleria delle carte geografiche, 1580–1585) a gallery decorated with 40 maps-murals that cover a length of 120 m. Impressive topographic details, perspective, plasticity, and rich colors have been used to tell stories related to the places: battles, naval battles, cities, and symbols of power. The application of light shading and color gradation for the representation of the relief of the land and the waves of the sea is remarkable, as well as the use of a realistic dark blue for the depiction of the sea. Colors and symbols are in harmony with Michelangelo's paintings that adorn the Capela Sistina to which the gallery leads. The inherent characteristics—harmony, colors, and plasticity of forms—of the mature Italian Renaissance period are predominant, but these maps are a unique case. It will take a long time for the technical means of creating maps to evolve, in order to continue this way of cartographic rendering. However, it should be noted that for the relief rendering, two different modes have been used, depending on the scale. On the larger scales maps, the tonal gradation has been used for the representation of the 3D relief, just like in the works of the representational painting. The impression of the relief, just like the plasticity of the figures, is achieved by the combination of tonal gradations, so that the one closest to the observer (higher altitudes) is lighter and the farthest (lower altitudes) more dark. On the smallest scales maps, the method of representation uses the expressive means of narrative painting. The land relief is represented in front view and the tonal gradations are used in relation to a light source: the light tones are located on the side from which the mountain masses are illuminated. Swiss relief rendering maps approach this style.

The first map printed by copper plate engraving, the map of the Mediterranean (Figure 4, adapted from [34]) by the Italian cartographer and engraver Paolo Forlani, was published in 1569. A portolan, but without following the previous structure and content of the maps of this category where color is used to represent political or geographical divisions and the corresponding pictorial symbols that prevailed in the previous two centuries have been eliminated. The countries and the continents are depicted by the inscription of their name, with color along the inner side of the coastline of some of them and by coloring their interior with a lighter color tone. The sea background is covered by sparse tiny black dots, but the coastline towards the sea is highlighted in blue. Its color sequence is in complete correlation with that of Paolo Veronese's painting (Figure 5, adapted from [35], Attribution: Paolo Veronese, Public domain, via Wikimedia Commons) as shown in Figure 6. Slightly warm colors of medium and low saturation: green, beige, ocher, orange, and blue. The differences in saturation, although relatively small, are expected due to the technical means of creating the two images, copperplate and oil painting, respectively, but also to the different aesthetic requirements of their shaping.

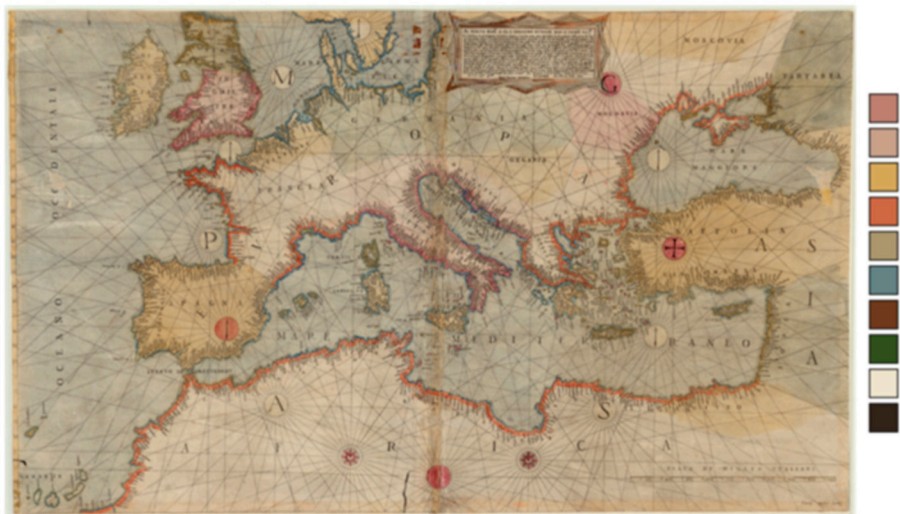

**Figure 4.** Paolo Forlani's portolan of the Mediterranean Sea and its color scheme.

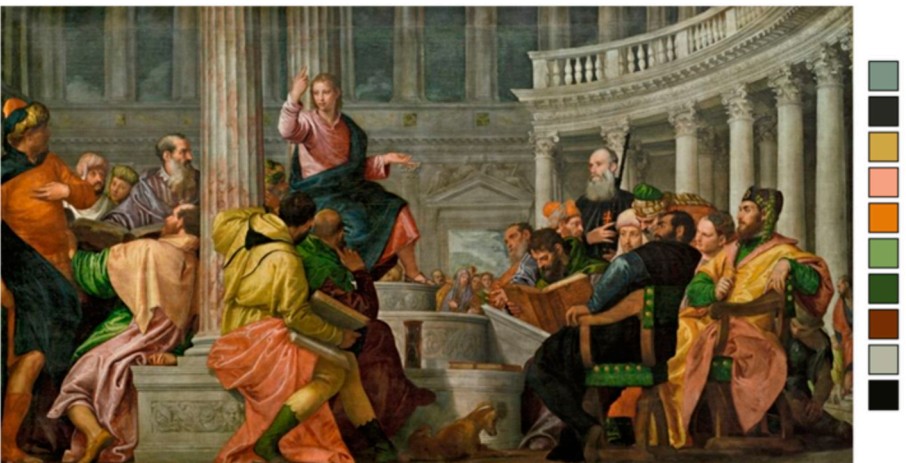

**Figure 5.** Paolo Veronese: "Jesus among the doctors at the Temple" and its color scheme.

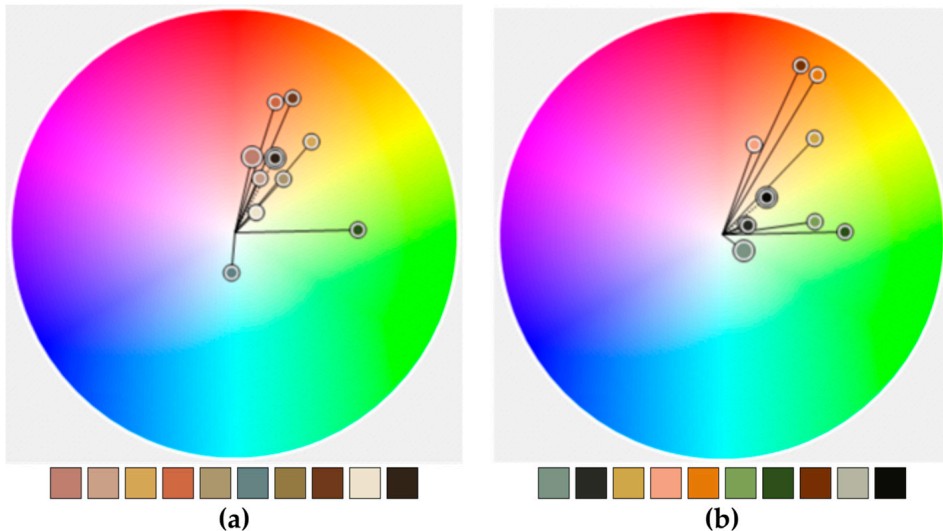

**Figure 6.** Color analysis and comparison: (**a**) Paolo Forlani's portolan (**b**) Paolo Veronese's painting.

Despite the gradual changes in maps in the late 16th century, the portolan atlases continued to emphasize the interest in depicting the world as new landscapes were discovered. Thus, the work of the Portuguese Fernão Vaz Dourado is very popular. Both Atlas [36] and Atlas with Portolans of the Old and New Worlds [37] (1568–1580) reflect a period when map features had to be visually impressive, on the one hand, and useful in navigation, on the other. These atlases are attributed to Dourado, based on their illustration and design style. Apart from the descriptive richness of these maps and the semiology of the design, the vivid colors of the illustration and many intricate design details give them great visual power (Figure 7). The colors of the symbols of power, blue, red, and gold, bring dominance to the foreground. Crowns, fleurs de lys, flags, and coats of arms pop up in the foreground of the image, forcing the conquered geographical area—both on land and at sea to retreat as a vassal in the background. The influence of Renaissance painting is clear: the color combinations (green, red, blue, gold, and earth tones) give naturalness to the objects, but also intensity and reason for existence in the decorative elements, which support the whole. The maps are based on color, just like many paintings of the Italian Renaissance and Mannerism. Examining the color scheme adopted by El Greco in his work "Christ healing the blind" (1570) [38] and comparing it with the corresponding map, it becomes obvious that both cover the area from the end of the warm to the cold blue through the green, have medium and low saturation values, and have a wide common section (Figure 8).

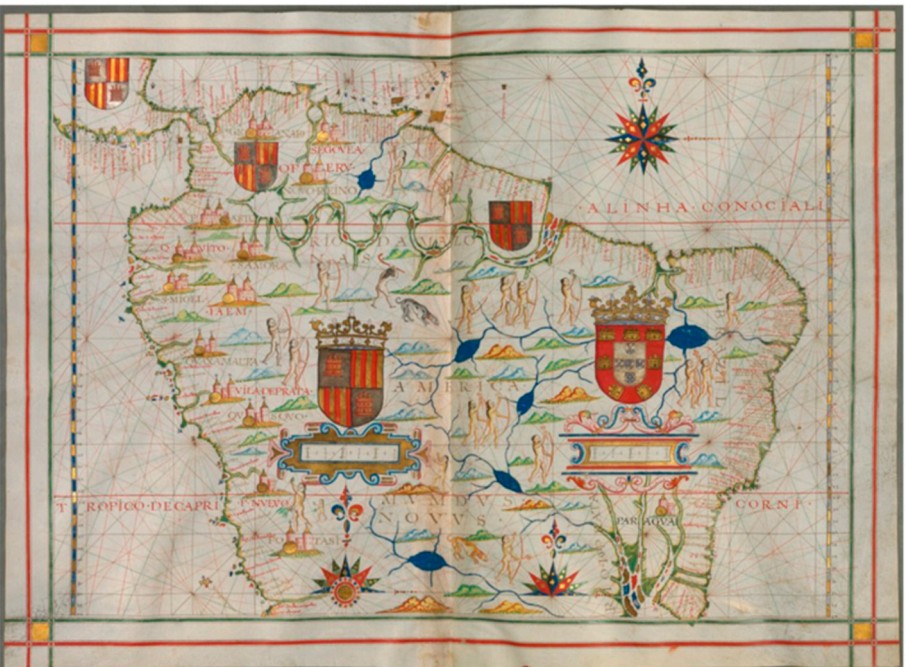

**Figure 7.** Fernão Vaz Dourado: The Atlas of Naval Maps of the Old and New Worlds.

The Guillaume Le Testu World Atlas ("Cosmographie Universelle selon les Navigateurs") [39] published in 1555, is another excellent specimen of 16th-century manuscripts and a typical example of the Dieppe School of cartography. It includes nautical charts that follow the style and content of the portolans as it had been established since the 14th century, but has been created with a strong painting visual character. This atlas, in addition to maps of individual areas, also includes six (6) depictions of the Earth, called "Projections". In all maps, the land part is covered by a green color that has a "tonal" gradation from inland to shores, while the small islands are colored red and some larger ones with ultramarine blue (lapis lazuli). Mountains are also designed in green, while rivers and lakes in gray. The sea is covered with low saturated light blue and brown wavy patterns, except for the Red Sea which is depicted in the "Sixth Projection" map and is drawn with a red line. Ships and sea creatures or monsters sail in the oceans, and symbols of royal power

and sovereignty are visible on all maps of this atlas. The background of the sky in medium blue harmonizes with the corresponding green background of the land, with lighter clouds and winds painted as male faces. The toponyms are written in bold font in red and gray. The color schemes that have been adopted are mostly cool (Figure 9) and harmonious in terms of the value of the colors. The high saturation is used to emphasize the symbols of power, which mark the domination of the lands. The atlas' color scheme is integrated in the context of the appearance of the Italian Renaissance's paintings.

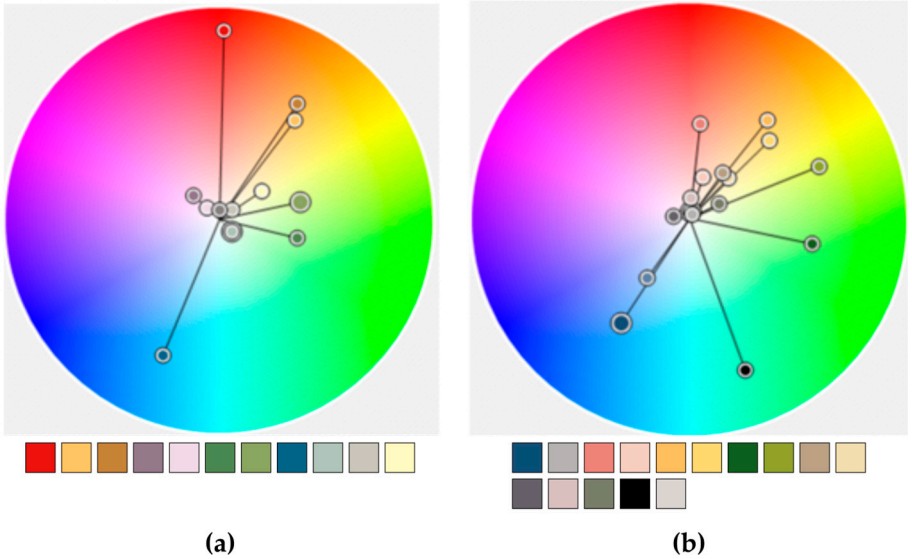

**(a)** **(b)**

**Figure 8.** Color analysis and comparison: (**a**) Fernão Vaz Dourado maps (**b**) El Greco painting.

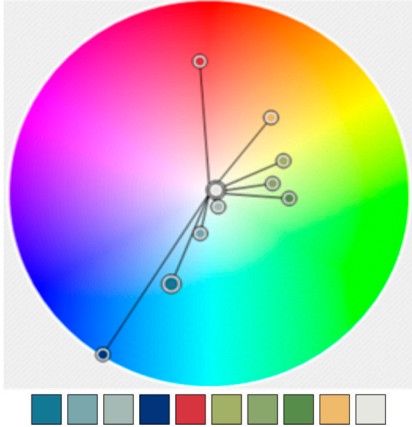

**Figure 9.** Guillaume Le Testu World Atlas color analysis.

The last quarter of the 16th century is marked by the Flemish cartographers Gerardus Mercator and his almost contemporary Abraham Ortelius. Mercator maps were sought not only for their content, but also for the italics fonts used to indicate toponyms. The Mercator Atlas [40] (Figure 10) published as a whole in 1595, is a combination of three sections published in 1585, 1589, and 1595, respectively. Mercator's maps signal significant changes: emphasis is placed on the geographical space and the use of pictorial elements is limited to the use of frontal aspects for the rendering of the mountains (with brown color and light shading that gives them a highly dramatic character), forests (green), and castles/cities (red). The settlements are symbolized with a small circular symbol and are accompanied by the inscription of their name. The colors chosen in this atlas (orange, beige, green, and yellow) have a strong visual affinity with the colors chosen by Pieter Bruegel the Elder [41]. These are low and medium saturated colors, which are placed in almost

one third of the color wheel (Figure 11), on the warm side, except for the blue-gray which marks the sea beyond the coastline, which is placed in the cool side.

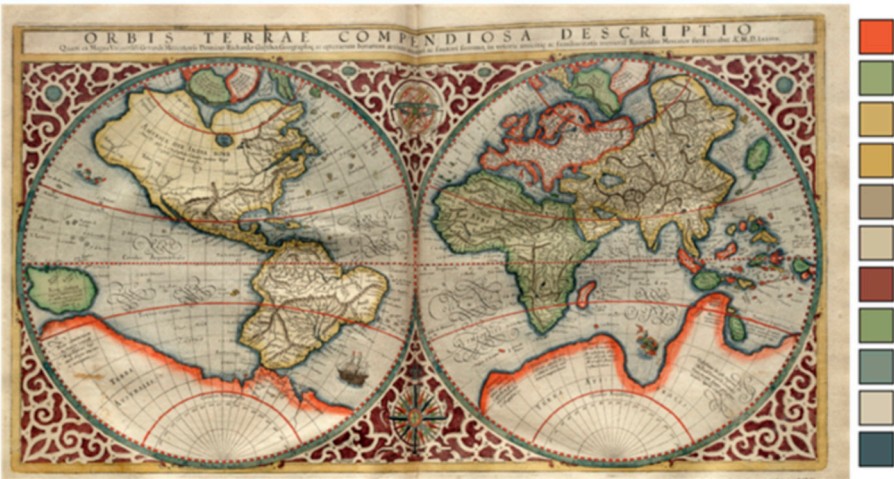

**Figure 10.** Gerardus Mercator Atlas and its color scheme.

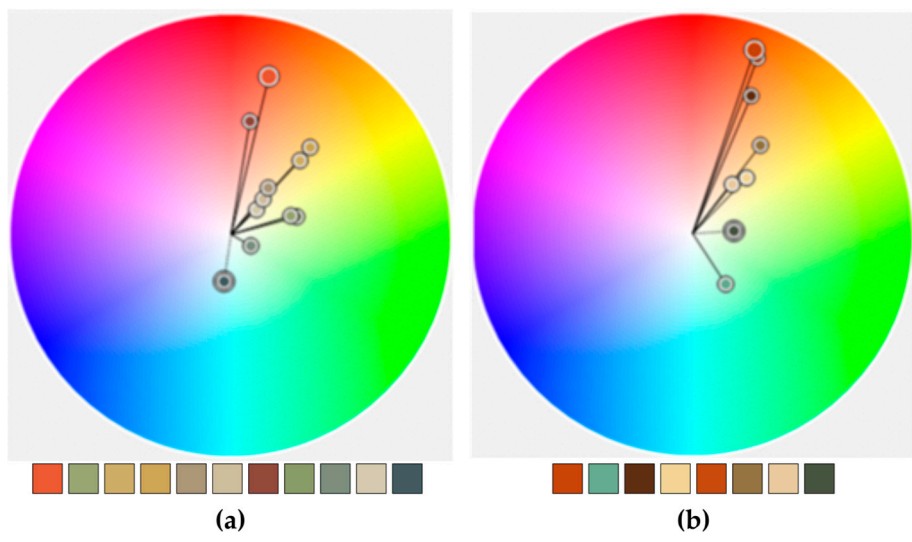

**Figure 11.** Color analysis and comparison: (**a**) Mercator atlas (**b**) Pieter Bruegel the Elder painting.

### 3.3. The Era of Baroque

In the golden age of the Dutch discoveries, the need to demonstrate the power, sovereignty, and grandeur brought about by land conquests determined the character of maps: magnificent compositions with complex visual character and rich marginal deco-ration. As in many paintings, the elements surrounding the central theme are not related to it, but are spectacular and they often have a strong allegorical character. Observing the overall appearance and structure of maps of that period, one can see the complexity of its composition. Geographic information is complemented by elements of astronomy, history, and mythology. The density of the image refers to complicated compositions, characteristic of the painting movements of the late 16th and early 17th century, the transition from the Renaissance to Mannerism and Baroque. The controversy of "form" over "color" continued with the controversy of the beauty of forms over "naturalism". The Flemish painter Peter Paul Rubens, despite being in Rome, which was the center of developments in painting and discussions around it, maintained the tradition of Flemish painters who focused on the realistic rendering of visible reality, but also on the emphasis on detail. In his works, the elements of the painting composition are crowded around a central object; there is a

wealth of light, a variety of textures and great vibrancy that contribute to the emergence of the subject, which is often allegorical. His artistic style belongs to the Baroque period and is characterized mainly by large, impressive compositions that tend to grandeur.

The aim of the Baroque style is to impress through intricate designs and luxurious decorative elements that shape and dramatic and theatrical style add imposingness to the work. These features are also found on Willem Janszoon Blaeu's map, as well as on other maps of Flemish, Dutch, and German cartographers of that period (e.g., Willem & Johannes Blaeu, Jodocus Hondius, Henricus Hondius, Mattias & Johann Bussemachaer, Joan Blaeu etc). Color not only has a decorative role, but is used to distinguish states or continents or individual elements (compass roses, etc.). As in the Baroque artistic style, the striking effects played an important role and intricate decorations defined the painting itself, the same way with the maps of the Baroque era, the constant addition of decorative elements made the maps intricate and striking, emphasizing the fringe elements. This fact adds to the maps the prestige and the glamour of the murals (religious paintings) with which churches and monasteries are decorated. The striking details of the physiographic elements and the intricate fonts are added to the intense painting decoration of the margin of the map, give an exuberant character, and are in line with the characteristics of this artistic movement that was definitively formed until the first half of the 17th century. The overall visual impression of these maps is characterized by color harmony between the main theme and the peripheral decoration. Vibrant or hazy colors of medium or high saturation, add intensity, luxury, and visual power. These are the characteristics of Baroque colors and are recognized not only in the masterpieces of painting, but also in the maps of the same era. From the study of maps and paintings of the Baroque period, the results of the color comparison are presented selectively in Figure 12 for one map made by Blaeu [42] and one of Rubens' painting [43]. The similarity of the colors is confirmed, both for the hues and the saturation range. Of course, due to the different technical means, the highest saturation values in the painting are expected.

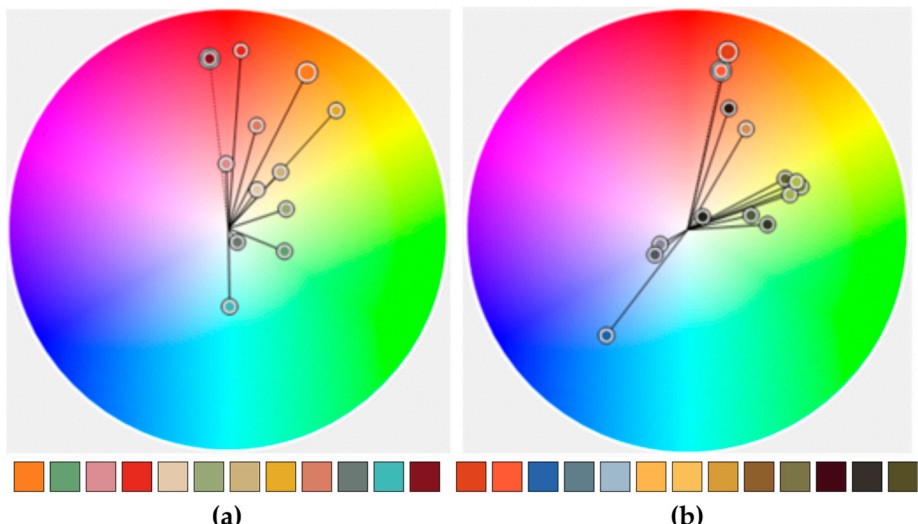

**Figure 12.** Color analysis and comparison: (**a**) Blaeu map (**b**) Rubens painting.

### 3.4. The Trends during the 18th Century

As part of the transition from Baroque to Romanticism, the marginal decorative elements are gradually eliminated from the maps, or limited to the cartouche. Their form gradually changes, overcoming the intensity and complexity of the Baroque and going to the grace and kindness of Rococo, as reflected in the paintings of the early 18th century in France, but also in Italy, England, Germany, and Austria. The paintings try to capture the pleasant everyday scenes, with idyllic landscapes and people from the aristocracy in various occupations, going beyond the strict limits imposed by the church. The colors are

soft and transparent with pastel tones, without deep shadows. They try to give the works a graceful, light tone and elegance. The decoration acquires a character of subtlety that expresses the taste of the French aristocracy and chivalrous grace [4]. Leading painters are Jean Antoine Watteau, Jean Honoré Fragonard, François Boucher, Giovanni Battista Tiepolo, and William Hogarth. Scientific developments, such as astronomical discoveries and the measurement of longitude, reinforce the autonomous identity of the map and weaken the need for decoration. An excellent example is the world map of Guillaume De L'Isle maybe the most important French cartographer of the time. Based on the-then-recent astronomical observations, not only did he contribute to the recalculation of latitude and longitude, but he incorporated this information into his maps, thus changing the accuracy of the maps. The change that is taking place in the appearance of maps in Northern Europe is great for the conditions of the time. Even the maps of Dutch cartographers with the heavy heritage of intense decoration, have replaced, in whole or in part, the decorative themes with cartographic inserts, visibly lightening the image and removing the dramatic and imposing elements. The emphasis is on the depiction of the geographical space, as, in the subjects of the paintings, the emphasis is on calm and carefree scenes, without action, without intensity. Colors are used without the chromatic visual strength of colors used in the 17th century; they are duller, lighter, as in the works of Antoine Watteau. A selective example of the similarity of the colors between maps and paintings during the Rococo period, is presented in Figure 13 about the comparison of the 1721 map by John Senex [44] and the painting of Antoine Watteau "Pilgrimage to Cythera" (1717–1719) [45]. The colors belong to the same area of the color wheel, so they have a common range of mainly cool shades and even with similar values of brightness and saturation.

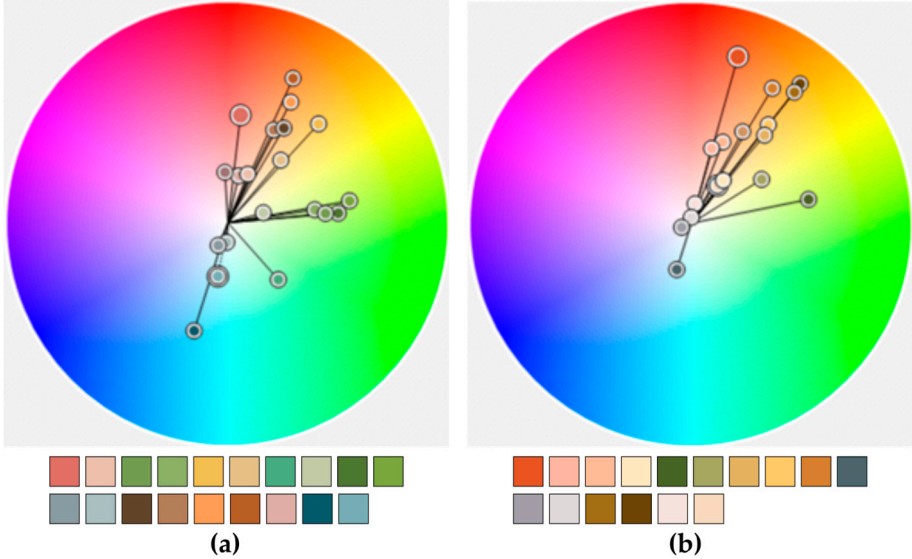

**(a)** **(b)**

**Figure 13.** Color analysis and comparison: (**a**) John Senex map (**b**) Antoine Watteau painting.

Neoclassicism succeeded Rococo, a style that embodied grace, asymmetry, and enhanced appearance with the use of elegant and delicate decorative elements. It was developed in Europe in the second half of the 18th century and relied on symmetry and harmony, virtues of ancient Greek and Roman art, bringing them back to the Renaissance. The works of the artists should be imbued with the "gentle simplicity and calm grandeur" of the works of classical antiquity. Neoclassicism opposes the exaggerated character and complex expression of Baroque, as well as the lightness and division of the Rococo form. Neoclassicism painting is characterized by the idealization of forms, clarity, and emphasis on outlines, the superiority of design over color, the accuracy and limited use of detail, and the avoidance of depth rendering. The forms become strictly descriptive and, due to the clarity of the outlines, they emerge from the painting like sculptures. The shadows

and fluctuations of the lighting—a strong element of Baroque painting—are constantly softening and in some cases completely disappearing. The viewer is obliged to focus his attention on the first level of the work and usually on the center, where the most important part of the composition is located. The rigor of Neoclassicism pervades the 1784 map of the Homann Heirs publishing house [46]. The map is located in the center of the observer's attention. There is no margin decoration; there is information in the form of text and tables instead. The colors, yellow, orange, green, and red, are also found in the work of Benjamin West "The Death of General Wolfe" (1770) [47].

*3.5. The First Half of the 19th Century*

In the context of the industrial revolution in the 19th century, mass mechanical production dominated over hand-made creation and production. During this period, artistic creation shifted more and more to the capitalist bourgeoisie and national academies, with merchants and art critics gaining a strong foothold. Paris had become the artistic capital of Europe, with the French Academy and official painting exhibitions defining developments in art. It is in this context that the transition from Neoclassicism (which began in the second half of the 18th century) to Romanticism takes place. In the field of cartography, the 19th century is characterized as the "Age of cartography". Cartographic creation and production in the 19th century acquired characteristics of professional specialization and the gradual standardization of maps began internationally, mainly through the establishment of official state cartographic organizations and institutes. With the development of lithography and the implementation of color printing, it became feasible to reproduce a large number of copies, which was used extensively to disseminate the map for economic, military, and educational purposes. The methods of map production and reproduction inevitably affected its style. Uniform cartographic practices and topographic maps were gradually standardized. In art, during the first half of the 19th century, Romanticism prevailed. Its main feature is the emphasis on evoking strong emotion through art, as well as the greater freedom in form, compared to most classical concepts. In Romanticism, the dominant element is the emphasis on emotion, not so much against logic as against its one-sided domination. Emotion, imagination, and lyricism are opposed to logic and pettiness. The colors are rich, the outline is slimming, the composition is full of movement and energy and the touches are free. Intense and contrasting movements and dramatic shading are some of the main features of this art and are very reminiscent of Baroque art. Typical are the artistic work of: Goya (Spain), Turner (England), Géricault and Delacroix (France), and Friedrich (Germany). Romanticism opposes the effects of the Industrial Revolution which led to the decline of handicrafts and replaced handicrafts with machine production. At the same time, a new middle class was developed that lacked tradition. Landscape painting, in the context of Romanticism, is part of the artist's freedom to choose a subject. Turner is considered one of the greatest landscape painters of the 19th century; he was a member of the Romantic Movement and became known as "the painter of light". Gilbert's world map (1839) [48] shows the use of colors similar to Turner's works, but it is not possible to ignore the aspects of the terrain at the base of the map and not to relate them to the imposingness of the visual scene of Caspar David Friedrich's artistic work, especially the emblematic painting "Wanderer above the sea of fog" (1818) [49]. The same approach is observed in the world map of Alexander Keith Johnston (1854) [50] and in the impressive Natural Atlas of Johnston & Humboldt (1850) [51].

Romanticism was overthrown in France by Realism. In painting, Realism rejects the emotional tone of Romanticism and advocates the depiction of real scenes in a realistic way, without a mood of embellishment. The leading Gustave Courbet, Jean-François Millet, Honoré Daumier, and Jean-Baptiste-Camille Corot are important painters of this movement, which had an impact and influenced artists in other countries, such as Britain, Germany, and Russia. The hazy colors of the paintings of this period are characteristically found on the map of Europe, which was published in London on 1861 [52].

### 3.6. The Second Half of the 19th Century and the 20th Century

Synonymous with the 20th century, the philosophical and artistic movement of Modernism is characterized by the spirit of opposition and deconstruction of conventional ways of thinking, expression, and representation. Modernist movements overthrow what is considered outdated or inappropriate in the new environment of a fully industrialized world. In painting, Modernism undoubtedly began with Edward Manet, but its peak is evident from the beginning of the 20th century until 1930, while its existence continues until Postmodernism.

One of the main artistic movements that used color as the main expressive means of individual and artistic expression is Impressionism. For the Impressionists, the characteristic of the visual scene is the light, which determines the impression, totally and partially. The light creates areas that are recorded with specific local color characteristics, without keeping the form unchanged, that is, the shape of the object. On the contrary, the fragmentation of the structure in combination with the special character of the colors, with emphasis on the primary shades, but also a pastel overall impression, realized with small visible touches, are special features of this artistic movement, the main means of expression of which is color. Another feature, which is found in several works, but mainly in those of Renoir, is a diffuse haze that is mainly due to the lack of outline, but also to the tendency to implement in the work the way in which a visual scene is perceived by the human visual system: everything in the central field of view is clear (focused), while what is perceived by the peripheral vision is blurred. Emblematic figures of this movement, apart from August Renoir, are Claude Monet, Edgar Degas, Camille Pissarro, and many other painters associated with it, without being considered its exponents, as either they had more influences by other movements, or the main character of their work (their identity) is part of later artistic movements. In Edward Stanford's color map [53], the main visual characteristics of the colors brought to the painting by the Impressionist painters are present. Despite the differences in the coverage area, on the map the areas of colors are larger, while in the works of the Impressionists the colors occupy small fragmented areas, which makes color analysis difficult, a comparison can be made with Monet's work [54]. The color sequences of the map and the table are placed in close areas of the color wheel, they cover a similar area in it, that is, they are similar shades; they have low saturation and high brightness (Figure 14).

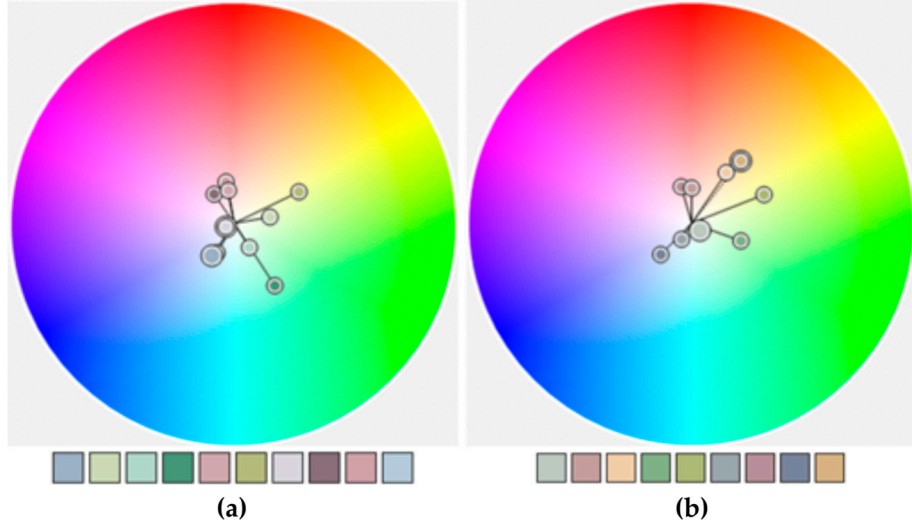

**Figure 14.** Color analysis and comparison: (**a**) Stanford map (**b**) Claude Monet painting.

The end of the 19th century finds artistic movements succeeding each other at a speed that would have been unthinkable a century or two ago. Impressionism handed over the baton to Post-Impressionism, which developed from 1886 to 1905 and that, in turn,

to Fauvism. The Post-Impressionists reacted to the fragmentation of the theme structure used by the Impressionists and used color to reinforce form, with no interest in naturalness. With Cézanne at the forefront, the form began to approach its geometric simplicity, a path that would lead modern art to abstraction. Despite the lack of common beliefs, post-Impressionist painters, each with their own unique character, used color in a more flat way. The deep Baroque light and shades contrast passed into the color gradations of Impressionism and began to become flat in the Post-Impressionism, Fauvism, Symbolism, and Expressionism that would follow. Paul Cézanne used a lot of burnt orange, blue, and bluish green, medium and low saturation colors in general [55,56]. Vincent Van Gogh is famous for the sulfite yellows he combined with the blue and the characteristic brush strokes of the shading, widely used in engraving. Paul Gauguin is famous for the use of complementary colors. The Fauvists, led by Henri Matisse, brought the bold color combinations, but at the same time established modern art. In various editions of Johnston W&AK Atlas [57], the use of green and burnt orange at various brightness levels recalls Cézanne's color vocabulary. The colors of [55,57] are located mainly in the cool part of the color wheel, except for the orange, with low brightness colors and medium and low saturation (Figure 15).

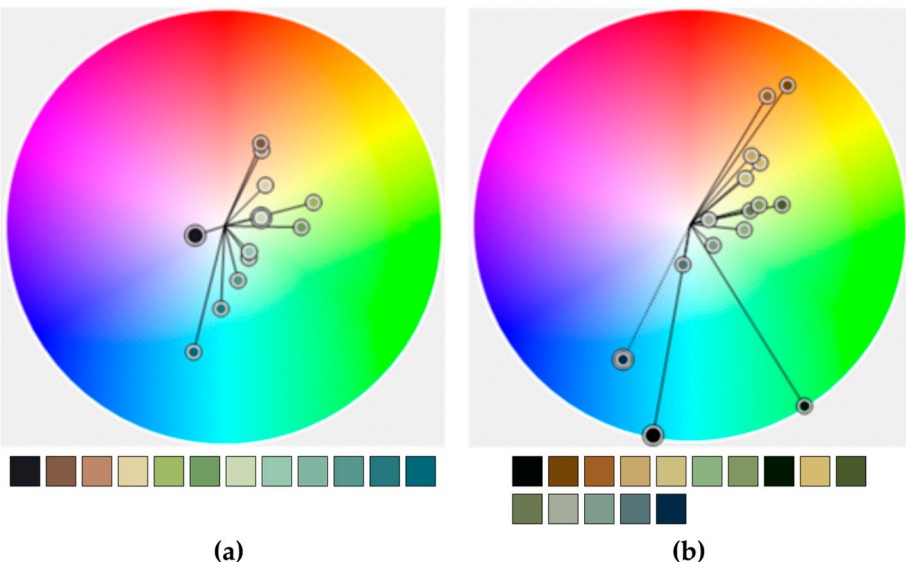

**Figure 15.** Color analysis and comparison: (**a**) W&AK Atlas (**b**) Cézanne painting.

The color map of 1897 [58] was published by the French publishing house Didot-Bottin during the Belle Époque, and is clearly part of the Art Nouveau with the characteristic curved decorative elements, inspired by plants and flowers, the flat colors that prevailed in commercial posters, and also in paintings (Toulouse Lautrec, Gustav Klimt, Alfonse Mucha etc.). In the map [58] and in the Alfonse Mucha poster [59], the colors occupy a very narrow area of the color wheel (Figure 16); they are earthy beige, hazy green, and brownish red. A similar comparison can be made between the Le Petit Journal map [60] and other Mucha posters.

In Expressionism the artist primarily expresses himself/herself and the visual medium for this is color, without form being so important, thus establishing the abstract art. Pioneer of abstract art Wassily Kandinsky studies, uses, and teaches color and transcends physical experience giving weight to its spiritual dimension. The point is that Fauvists and Expressionists have one thing in common: bold, strong, opaque color [61,62] and will influence not only paintings, but also graphic works in general. The map of climate zones [63] is in the spirit of Wassily Kandinsky's color choices. High saturation, high brightness for colors mainly warm, but also with the presence of some cool ones, all placed in the same part of the color wheel (Figure 17).

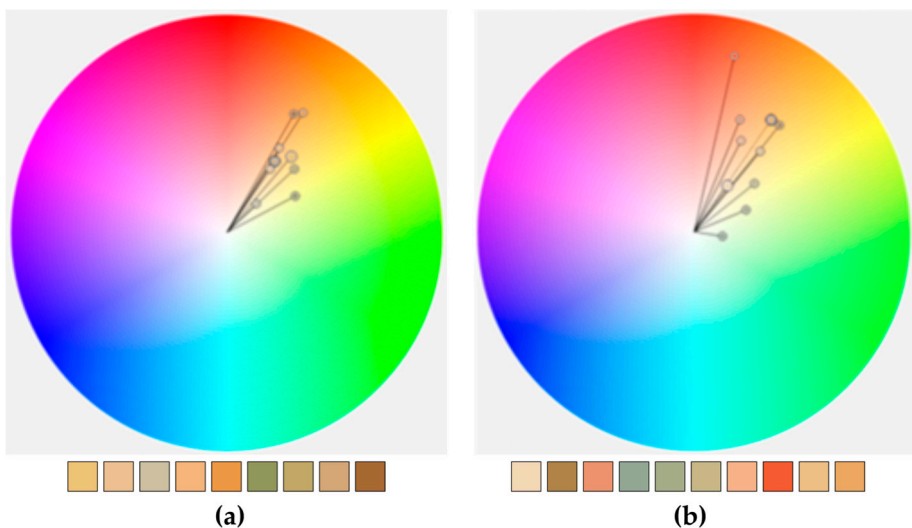

**Figure 16.** Color analysis and comparison: (**a**) Didot-Bottin color map (**b**) Mucha poster.

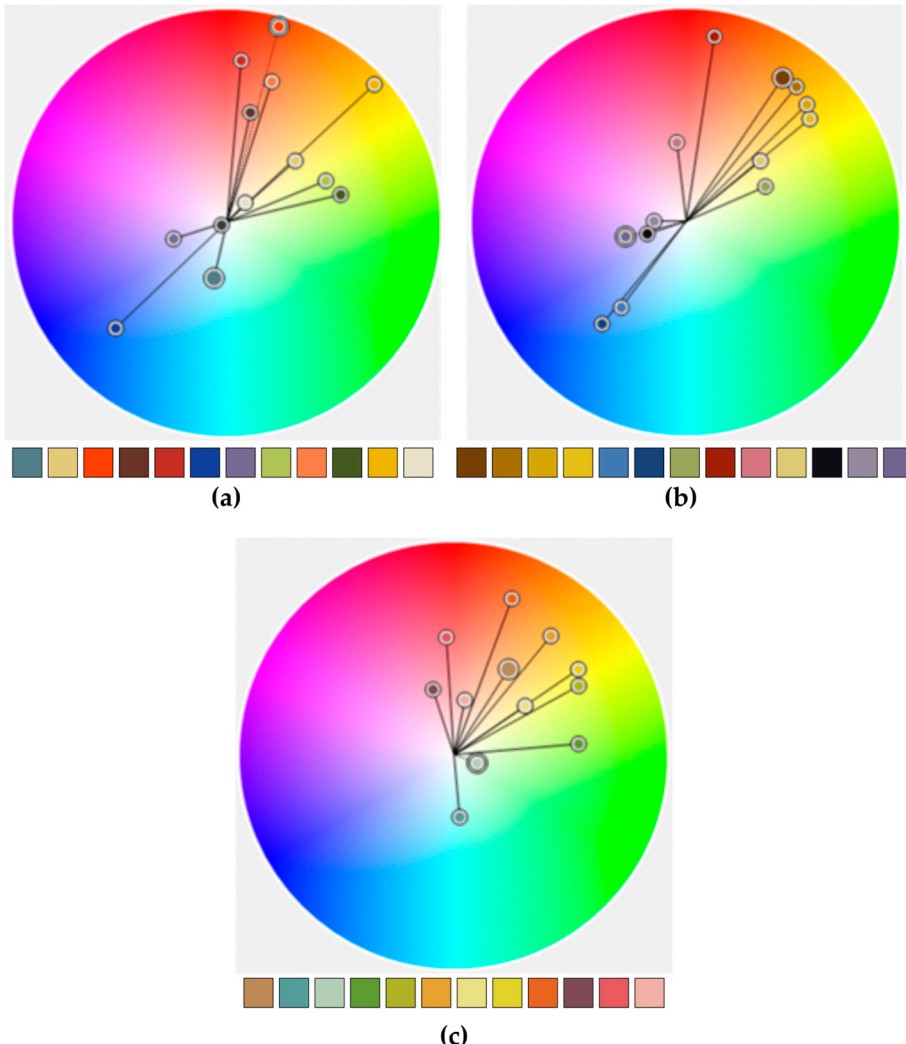

**Figure 17.** Color analysis and comparison: (**a**,**b**) Kandinsky paintings (**c**) Climate zones map.

The top moment (specimen) of Modernism in cartography is the map of Henry Beck, which is a landmark for the topological depiction of geographical information [64]. Chrono-

logically it belongs to the era of Art Deco style with influences from the abstraction and geometry of Cubism, and the bold and vivid colors of phobia. Piet Mondrian had similar influences and since 1915 he has been experimenting with abstraction, which he conquers by maintaining only simple geometric shapes and vivid colors in his painting discourse, developing the non-representational form of Neo-Plasticism [65]. After World War II, topographic maps of official national cartographic agencies are produced in accordance with the specifications and instructions issued by NATO through STANAGs. World maps are usually issued for educational purposes and are contained in school atlases: political, geophysical, and thematic. However, the value of the influence of Modernism is not degraded as the colors are vivid and flat and the abstraction has been assimilated not only in the content, but also in the form of the maps [66,67].

### 3.7. Early 21st Century

At the end of the 20th century, cartography, or better cartographic creation and production, is described by the terms digital cartography or automated cartography, both indicative of the complete use of computer systems in the cartographic process. After all, technical means have always played a determining role in the implementation of the map. Equally sweeping is the use of computers, software, and multimedia in artistic creation. If photography, with its entry into museums and art galleries and its integration into the Fine Arts, began to undermine the foundations of painting as one of the most powerful and creative artistic means of expression, digital photography, the creation of images using computers, video art, and installations, are just some of the contemporary artistic activities, which a few decades ago it was unthinkable, if not rejected, to be considered art. In the field of cartography today, the use of maps and geospatial data is adopted by the average internet user. Online maps are published either by official National Cartographic Agencies (e.g., IGN-France), or by private companies (e.g., Google) and voluntary crowd sourcing data actions (e.g., Open Street Map) etc. The online map services are widely used for locating points of interest, navigation, planning, and monitoring of trips, real-time information (e.g., traffic load), training, etc., but also as a background for the creation of (new) maps. They provide global coverage and are used by people all over the world. These services have a huge impact on the millions of Internet users, giving them a (realistic) view of the world. Competition between providers results in a continuous improvement of these products, but the most important thing is that nowadays, more than any other time in the history of cartography, the map itself is used daily at the same time and for different uses by people around the world. Nowadays, in the age of digital communication, the map as a medium is becoming more widespread in society. Never before were maps directly accessible by the user, never before were maps an integrated part of everyday life, and this is a huge achievement. It is the most unified and integrated means for simultaneous representation of position and descriptive information in textual form. The enthusiasm that comes from the ubiquitous online map service should not be a narrow self-referential limit, but should be matured with choices, utilizing the principles and heritage of cartography [68].

## 4. Discussion

A map is not a typical commercial graphic product. It conveys information about the geographical area, which has measurable and qualitative value and the signature of the body (or the individual) of its creation. It has the status of a symbolic image of geographical reality and with paintings they have common elements: image, symbolization, representation. Of course, the map lacks the freedom to choose the subject and the form of the paintings, but, over time, it has followed or assimilated the choices in the use of color as it emerges from the prevailing artistic trend.

From the Gothic rhythm and the exclusive influence of the Church to the era of digital technology domination, the journey of color in painting clearly influenced the application of color on maps. In many cases the visual comparison results in strongly similar visual impressions in such an eloquent way that you may not need color analysis

and comparison in quantitative terms. However, the similarities are documented by creating the corresponding color sequences and comparing them with quantitative terms captured in the color wheel [20].

The paintings and maps that were selected and examined in detail and their subset presented in this paper, support the impact of painting, without implying that this trend was intentional. Apart from the cases of collaboration of cartographers with painters, painting is part of the intellect of a place and an era and shapes the culture as well as the aesthetic approach—among other things—of graphic products. Through art, an attitude is developed, familiarity with color is achieved, and a number of images that serve as a source of inspiration can support the adoption of artistic expression or the creation of a style.

The apparent distancing of cartography from art over time may be related to scientific knowledge, measurements, and the evolution of technology that made it possible to create and produce maps without collaboration with artists (for the design and/or coloring). However, it is not overlooked that not only cartography, but also painting gradually assimilated scientific knowledge and incorporated it into the means of expression of artists. A prominent example is Georges Seraut, the founder of Neo-Impressionism, who applied the trichromatic theory of color vision to his personal painting style [69]. Is this personal style related to color photolithography and the use of CMYK separation screens at a specific angle for each color of the four color printing method [70]? In addition, the application of perspective in the works of the Renaissance and its rejection by the Cubists in the shaping of their work connect art with science.

The power of the map lies in the application of cartographic symbolization and, here, the relationship with the precepts of art is inseparable. The foundation of the application of a more objective symbolization, Bertin's main visual variables [21] and, in particular, the three of them related to color, come from the standardization of color in painting and have exactly the same use that they have in paintings: the shade has the character of qualitative distinction, while the brightness (value) and the saturation a quantitative one [71]. Undoubtedly, the standardization of maps from the end of the 19th century created the need to establish rules, not only in the form (layout) and content of maps, but also in the application of cartographic symbolization in a less subjective way. This does not mean that a situation was created from scratch. On the contrary, the cartographic evolvement was formed based on the cartographic tradition and the integration of scientific knowledge, technological means, and the precepts of art. The heritage of art in cartography is important. The use of hue for the nominal differentiation of spatial entities and the use of brightness and saturation for quantitative differentiation [71] constitute a vivid paradigm.

The depiction of volume and plasticity of the three-dimensional objects in the relief representation, mainly in combination with the light shading, unequivocally substantiates this view. Here is the absolute application of the pop-up (warm) colors for the larger and the withdrawn (cool) colors for the lower altitude values, which are closer and farther from the observer, respectively [72]. The implementation of plasticity, i.e., the impression of volume, is created with the bright colors placed in the protruding areas. This technique was widely applied in the rendering of the plasticity of objects in paintings and during the evolution of the expressive means it was combined with the use of light and shadow [73].

The examples given above are part of a broader research and confirm the common path followed regarding the use of color in the maps and paintings of the respective periods. As is presumed through the experimental procedure followed, the color path in the maps tracks the color path in the paintings. From the works of the Middle Ages to the 21st century, the differences in color, both for aesthetic and for technical or practical reasons, are obvious. From heavy or luxurious colors with or without light shading applied even in complex shapes and compositions [74], the transition to flat colors and almost complete subtraction has taken place, as applied to the land-sea separation in the world online maps [75].

Until the middle of the 19th century the artistic movements used to change at a relatively slow pace, as did the style of maps. The influence of painting was then clearer as there was, on the one hand, collaboration of map makers with painters, but also more

time to assimilate artistic achievements or choices. The artistic movements from the middle of the 19th century were less and less lasting and there are cases of painters whose conquest of a style, a means of expression, was the trigger for the next experimentation, as if it was important for the personal artistic expression and recognition who will be the first to create a trend, who will be the first to conquer an achievement, a goal; to whom will the originality be credited? Certainly this has to do with the gradual weakening of collaborative movements and the prevalence of individualism and individual expression; the possible "anxiety" of acquiring a very personal and recognizable identity, an identity that differentiates the individual from others and makes him/her special. Individualism, the commercialization of art, the predominance of the intangible (in the form of digital) in all activities of modern culture, often leads to the questioning of traditional methods of artistic expression by characterizing them as conservative. Currently, this also governs commercial cartographic applications, as it becomes clear from the comparison of the most popular online maps [20,68]. The issues of commercialization, the commercial value, but also the authenticity of the works exhibited in the museums, concern the intellectuals in many ways.

It could be said that if today the map had to be part of the "philosophy" of an artistic movement or an art school, then it would be part of the Bauhaus School, which approaches design by combining functionality, simplicity, usability with an emphasis on color and the geometry of "form", and, most important, applying the precepts of art to utilitarian objects.

Paintings are a field of realization of the characteristics of color, a field of highlighting the relations of colors and the simultaneous contrast. Each of these can replace experimental research on how colors look next to each other and how color observations, affinities, and contrasts are perceived by the observer. Each of them can be a source of inspiration for creative cartographic composition, and if the institutionalization of specifications standardized the topographic maps, and in part the general-use maps, there remains room for creativity in the thematic maps, mainly in the digital environment of the internet, which today is the dominant platform for creating and disseminating maps.

## 5. Conclusions

This research was not based solely on the visual similarities between the paintings and the maps. Undoubtedly, the greatest power of color lies in the image. No word, no description of color can be as apt and accurate as the color itself. However, the analysis of the color sequences and their recording in the color wheel reinforces the initial subjective judgement, giving it an objective character, through the description in quantitative terms.

Despite, to a certain extent, the subjectivity in the selection and the reasonable assertion that may not be universally applicable, the findings are interesting and significant enough to form the basis for further research. It is noteworthy that in almost all of the comparisons made, the color sequences are placed in the same part of the color wheel and have almost the same range. Therefore, the color similarity between the paintings and the selected maps is supported both for the colors used and for the color character resulting from their saturation and brightness. It should also be noted that colors differ over time: the red, green, or blue of the Italian Renaissance has a different visual character than in Mannerism or in Modernism. El Greco's red is very different from Bruegel's red and foreign to Renoir, Chagall, or Mondrian's red. This variation of the color character is also recognized in the journey of color in the maps: the red used by Mercator is quite different of the one used by Dourado or Agnese. It should also be emphasized that the atlases which have been studied are characterized by a unified color approach, which gives aesthetic and artistic unity to the work.

It would be interesting if a similar research would be done outside of the Western European culture. For example, consider this relationship in Asian or Arab culture or, in a narrower context, in the 20th century, in artistic expression during the period of "actually existing socialism" in Eastern Europe. Japanese or Chinese culture, as well as the masterpieces of Arabic culture from the depths of time to the present day, are remarkable

and have left an important imprint on both art and cartography. The examination, for example, of Japanese paintings, has significant differences from those of the Western world. Japanese culture has a long tradition, among other things, in painting and design, with a strong minimalist approach. A glance at Sakai Hoitsu "Iris" [76] and Van Gogh "Iris" paintings, as well as a Japanese world map, is enough to deeply think about it [77]. It would be extremely interesting to consider further research on other symbol variables, especially for visually impaired people, as, in this case, any tactual perception cannot rely on color.

In conclusion, it is estimated that the initial objectives of this research have been achieved, not only through the experimental approach and the documentation of the respective conclusions through the listed color analysis, but mainly through the knowledge acquired during the study and analysis of the artistic and cartographic periods. The study of paintings and the ability to identify the respective period or the corresponding artistic movement based on how the color was used is an important resource that can be widely used, both in map composition and the evaluation of the cartographic result. The widespread use of online maps services is a great opportunity, but also a responsibility for cartographers not only to take advantage of the technological aspects, but to also integrate in cartographic practice and teaching the application of the precepts of art. In this way, maps can become as fascinating today as they have been in the past, without losing their contemporary character and functionality.

**Funding:** This research received no external funding.

**Institutional Review Board Statement:** Not applicable.

**Informed Consent Statement:** Not applicable.

**Data Availability Statement:** Images of maps and paintings were obtained from the Internet. Source links are mentioned in the References.

**Conflicts of Interest:** The author declares no conflict of interest.

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
