# Peer review of "Cartography and Art: A Comparative Study Based on Color"

_geographies, doi:10.3390/geographies2010007_

Round 1
Reviewer 1 Report
Dear Author, I will start with some general comments and then move to more detailed ones, referring to particular parts of the manuscript.
GENERAL COMMENTS:
The research tries to deal with an interesting topic of color similarities between maps and paitnings of different periods. The language used is appropriate and allows smooth reading.
Unfortunately, to me it lacks scientific soundness. The manuscript could form a part of a book (chapter) but not a scientific paper. My main complaint is the lack of methodology section. What is the background of analyses conducted? What do the color schemes tell readers?
Besides, the manuscript is hard to read due to the lack of appropriate referencing. The same applies to the figures. Are copyrights really an issue here? I assume you could include the works with proper referencing in the paper.
The selection of works for comparison look very subjective to me. How can readers know what is the background of examples selection? In conclusions section you state that maps and paintings from particular periods share the same color schemes but this statement is based only on some of the pairs of works compared. There is no numerical nor systematic visual analysis provided.
Unfortunately to me, the manuscript cannot be published at its current state. The author must consider preparation of an unequivocal methodology for his/her analyses that would raise scientific soundness of the paper.
I wish the author all the best and look forward to seeing further contributions in this area.
DETAILED COMMENTS:
-
Line 9: Please decide whether you want to call it a paper or study.
-
Lines 25-26: Could you please clarify this statement?
-
Lines 44-45: Reference should be provided.
-
Lines 48-71: I find this paragraph not suitable in its currect form. It lacks proper references and examples of the works/maps mentioned. It reads like an encyclopaedic section rather than journal paper. Please rewrite it.
-
Lines 50-53: Some example of such map could be useful.
-
Lines 107-109: I am missing such clear statements of the paper's goal in the abstract.
-
Figure 1: Figures and tables should be self-explanatory. In this color wheel, what do these lines mean? How about the color samples below? I see no point in introducing such figure without explanation of what can be seen on it.
-
Table 1: Where do color names come from? I imagine there might be 100 shades of Yellow within each color system. Why introducing this names instead of, for example, placing the color samples in the table?
-
Line 171: I suppose the author meant century AD?
-
Line 184: The term 'naif painters' should be explained. I also think that the spelling here is incorrect: Naïve/Naïf.
-
Line 336: What is a 'Sixth Projection"? Either explanation or a reference should be provided.
-
Lines 352-353: I don't understand this sentence. How is this beggining changing the maps?
-
Lines 421-423: There is a repetition >> 'The colors are soft and transparent with pastel tones' (...) 'The colors become softer and more transparent (pastel)'
-
Section 2.6.: I find this fragment non-chronological. It's a mix of 19th and 20th century.
-
Lines 697-698: How could author confirm this statement? The paper only includes selected examples without any numerical or even systematic visual analysis.
-
Lines 718-728: I really doubt that such sections are appropriate for this journal.
-
Lines 751-752: Perhaps, but at the same time, there might be numerous other pairs that would be placed on completely different parts of the color wheel. I am not saying that this is the case but readers lack proofs for your statements.
-
Lines 762-763: Yes, indeed. Another idea for future research is to consider other symbol variables. On tactile maps for people with visual impairments, color is not really useful.
Author Response
Dear Reviewer,
Thank you very much for your thorough review. I have taken into account your comments and remarks to improve the article . Please see the attachment for the responses.

Reviewer 2 Report
Very interesting study showing cartography as an art. Author tries to correlate maps (especially their colors) with the artistic periods in Western Europe dated from the end of Middle Ages to 21st century. More than 100 maps and painings. Both of them (cartography and art) have their scientific conections and inspirations.
A lot of examples of the arts inspired by maps – very interesting introduction to the study.
Paragraph started from line 72 to linie 97: please consider the reference to cartographic literature in the context of a model of map communication (the communication proces from the cartographer to the map user), it was discussed in main cartographic textbooks (Dent, Slocum, Robinson).
When discussing color issues, it is useful to refer to classic textbooks on cartography (Robinson, Slocum), and perhaps even to J. Bertin's Graphic Semiology.
Some technical remarks:
Line 6: Cartography or cartography, line 32: Science or science, line 34, 659: Art or art, line 656, 657: Painting or painting – are the capitalic letters really needed?
Line 124: the dot is needed at the end of the paragraph.
Author Response

(The authors gave the same response as above.)

Reviewer 3 Report
The article investigates the design of maps as a consequence of the dominant artistic movements and focuses on colour as an identifier of the corresponding artistic trend.
It takes general pictorial stages and for each of them observes the relationship between several paintings and maps. In doing so, they analyse and compare the colours used both in the selected pictures and maps, concluding that there is a colour connection between the paintings and the selected maps, for each artistic period.
The subject is original and undoubtedly interesting, since the aspect of the cartography that is produced cannot be separated from the visual culture of its time. And therefore the study of this relationship is necessary. Moreover, the article is impeccable in its writing, presentation, etc. and the discourse is perfectly spun.
However, I have two hesitations about recommending it for the special issue Geovisualization: Current Trends, Challenges, and Applications.
The first is due to the subject of the article: I am not sure how well it fits in with the geovisualisation aspects. The contribution is not aligned with the current topics on Geovisualisation. Perhaps journals such as Imago Mundi or e-Perimetron could be better options?
The second doubt relates to the methodology followed, which is not entirely clear to me, and concerns the criteria for selecting both the paintings and the maps to be analysed.
While reading the text, it's not completely clear to me how the previous selection work has been made; why are the ones mentioned in the paper chosen and why not others? What criteria have been followed to choose a particular painting and a particular map? I think this is not a trivial question and I am sure there are criteria that can be explained.
Folowing this idea, if other cases were analysed, would the results be consistent with the ones presented? More cases than those shown have been analysed, as indicated in the text; has not been developed an indicator of consistency expressing the 'validity' of the map - painting colour matches for each period? This issue is also not explained in the article and would be interesting to include, or at least to consider.
A minor detail; I think it would help the reader to divide the references in two, leaving the bibliography on one side and the painting/maps on the other. It would also help to include the links to those resources in the text.
Otherwise, I enjoyed reading this article, the subject matter is interesting and I think it is important not to lose sight of the fact that if maps are a product of the technology of their time, they are also a product of their artistic and social context.
Author Response

(The authors gave the same response as above.)

Round 2
Reviewer 1 Report
Dear author,
thank you for considering all of the reviewers' comments. I truly believe that this paper has improved significantly now, once it has been reviewed.
Although I would still like to see some more technical approach to this study, I don't want to hold it back any longer as it has potential for further development. Especially in the face of the fact that other reviewers found it suitable for publication.
For the future research, please consider some more controlled/parametrized way of qantitative evaluation of the paintings. Since you already have your dataset of more than 100 maps and paintings, you could just apply the designed methodology and extend you research in that way.
I wish you all the best in your research career.
The list of more detailed comments can be found below. Please refer to them before proceeding with publication:
-
Lines 69-70: Please confirm the correct way of referencing images - should the reference be placed in-text or in Figure caption?
-
Line 88: there's a typo.
-
Lines 107-109: Please confirm with journal's author guidelines the required way of direct citations.
-
Lines 129-130: Indeed, referencing Bertin in such research is rather compulsory!
-
Lines 166-167: Could you please elaborate more on these 'cartographic periods'?
-
Lines 185-186: What kind of metrics? The figures of colour wheels only provide readers with visual representation. There are no numerical measures mentioned. Please either describe them in more detail or just consider it a future reseach plan.
-
Figure 2: I am sorry but to me it's still unclear why these 10 colours are selected. Does the software look for the most common pixel colours across the whole image? This is the key information that all the comparisons are based on.
-
Line 221: What is the HSB tool? Is this some sort of a built-in tool within software used? I am unfamiliar with the software and thus it is hard to comprehend the methodology.
-
Line 249: I might be wrong but BC means to me 'Before Christ'. If author wants to be neutral, as far as I know 'BCE' would the correct term (Before Current Era) here.
-
Figure 5: I suppose the URL should be included in references. Here only the reference number should be placed.
-
Line 968: typo in 'judgement'
-
Lines 969-971: Okay, now it makes more sense. There is nothing wrong with such statement.
Anyway, wouldn't it be good to perform some sort of quantitative analysis? Could we get a similarity measure? Perhaps average distance between colours on a colour wheel that could be calculated for the whole dataset, instead of just visual evaluation of the selected examples?
-
Lines 992-994: If I could suggest something - as you decided to mention people with visual impairments, it would good to mention that tactual perception cannot rely on colour, which is the key in your current research.
Author Response
Dear Reviewer,
Thank you once again for your thorough comments and remarks, which I have taken into account to revise the manuscript. You will find all the responses in the attached file.
Your contribution to improving this article is crucial.
Kind regards,
Dr. Leda Stamou
